# BLESSING OF DIMENSIONALITY FOR APPROXIMATING SOBOLEV CLASSES ON MANIFOLDS

## ABSTRACT

The manifold hypothesis says that natural high-dimensional data lie on or around a low-dimensional manifold. The recent success of statistical and learning-based methods in very high dimensions empirically supports this hypothesis, suggesting that typical worst-case analysis does not provide practical guarantees. A natural step for analysis is thus to assume the manifold hypothesis and derive bounds that are independent of any ambient dimensions that the data may be embedded in. Theoretical implications in this direction have recently been explored in terms of generalization of ReLU networks and convergence of Langevin methods. In this work, we consider optimal uniform approximations with functions of finite statistical complexity. While upper bounds on uniform approximation exist in the literature in terms of ReLU network approximation, we consider the opposite: lower bounds to quantify the fundamental difficulty of approximation on manifolds. In particular, we demonstrate that the statistical complexity required to approximate a class of bounded Sobolev functions on a compact manifold is bounded from below, and moreover that this bound is dependent only on the intrinsic properties of the manifold, such as curvature, volume, and injectivity radius.

## 1 INTRODUCTION

Data is ever growing, especially in the current era of machine learning. However, dimensionality is not always beneficial, and having too many features can confound simpler underlying truths. This is sometimes referred to as the curse of dimensionality (Altman & Krzywinski, 2018). A classical example is manifold learning, which is known to scale exponentially in the intrinsic dimension (Narayanan & Niyogi, 2009). In the current paradigm of increasing dimensionality, standard statistical tools and machine learning models continue to work, despite the high ambient dimensions arising in cases such as computational imaging (Wainwright, 2019). One possible assumption to elucidate this phenomenon comes from the manifold hypothesis, also known as concentration of measure or the blessing of dimensionality (Bengio et al., 2013). This states that real datasets are actually concentrated on or near low-dimensional manifolds, independently of the ambient dimension that the data is embedded in.

In this work, we explore the consequences of the manifold hypothesis through the lens of approximation theory and statistical complexity. For a class of functions with infinite statistical complexity, we consider a nonlinear width in terms of how well it can be approximated in $L^p$ with function classes of finite statistical complexity. We consider how difficult it is to optimally approximate classes of functions with functions of finite statistical complexity in terms of $L^p$ distance. In particular, Theorem 3.1 demonstrates that on a Riemannian manifold, the optimal error incurred by approximating a bounded Sobolev class using function classes of finite pseudo-dimension can be lower bounded using only the *implicit* properties of the manifold.

### 1.1 RELATED WORKS

We review some literature surrounding the manifold hypothesis, including theoretical results derived from the manifold hypothesis, and lower bounds on statistical complexity required to approximate a function class. We note that the manifold hypothesis is sometimes replaced with the "union of manifolds" hypothesis, where the component manifolds are allowed to have different intrinsic dimension (Vidal, 2011; Brown et al., 2022). For estimating the intrinsic dimension, we refer to (Pope

et al., 2021; Block et al., 2021; Levina & Bickel, 2004; Fefferman et al., 2016); for representing the manifold or dimension reduction, we refer to (Lee et al., 2007; Kingma & Welling, 2013; Connor et al., 2021; Tishby & Zaslavsky, 2015; Shwartz-Ziv & Tishby, 2017).

Bubeck & Sellke (2021) shows that for a class of Lipschitz functions interpolating a noisy set of samples, if the empirical risk is below the noise level, then the Lipschitz constant of $f$ scales as $\sqrt{nD/p}$, where $n$ is the number of samples, $D$ is the ambient dimension, and $p$ is the number of parameters. Gao et al. (2019) shows that any class of functions that can robustly interpolate $n$ samples has VC dimension at least $\Omega(nD)$, and demonstrates a strict computational increase required for robust learning. Bolcskei et al. (2019) show lower bounds for the connectivity and memory requirements of a deep neural network for approximating function classes in $L^2(\mathbb{R}^d)$.

Chen et al. (2019) provide approximation rates of ReLU networks for Hölder functions on manifolds based on the width, depth, and total parameters, albeit still depending linearly on the ambient dimension of the model, assuming isometric embedding in Euclidean space. They provide approximations based on partitions of unity and classical constructions on near-Euclidean charts. The same authors provide associated empirical risk estimates and generalization bounds for ReLU networks in (Chen et al., 2022). Labate & Shi (2023) consider uniform generalization of the class of ReLU networks for Hölder functions on the manifold, using the Johnson–Lindenstrauss lemma to work in near-isometry to Euclidean space.

On the unit hypercube, Yang et al. (2024) addresses the complexity of approximating a Sobolev function *constructively* with ReLU DNNs by showing an upper bound on the Vapnik-Chervonenkis (VC) dimension and pseudo-dimension of derivatives of neural networks based on the number of layers, input dimension, and maximum width. Park et al. (2020); Kim et al. (2023); Hanin & Sellke (2017) consider lower bounds for the minimum width required for ReLU and ReLU-like networks to $\varepsilon$-approximate $L^p$ functions on Euclidean space and the unit hypercube. We generalize this line of work by deriving lower bounds on the *statistical complexity* in terms of the *nonlinear width*, c.f. Definition 2.4, required to approximate Sobolev functions on *compact Riemannian manifolds*. Sobolev functions define a sufficiently expressive class of functions that can model many physical problems, while also having sufficient regularity properties allowing for functional analysis. This work thus considers the difficulty of modelling physical problems over structured datasets with simple function classes.

In Section 2, we formally introduce the concepts of pseudo-dimension and desired notion of the width of a function class, followed by some existing results relating complexity to generalization behavior. We also briefly discuss Riemannian manifolds and prerequisite knowledge needed for the main results. The main result is Theorem 3.1, with proof given in Section 3.

## 2 BACKGROUND

### 2.1 PSEUDO-DIMENSION AS COMPLEXITY

We consider a concept of statistical complexity called the pseudo-dimension (Pollard, 2012; Anthony & Bartlett, 1999). This extends the classical concept of Vapnik-Chervonenkis (VC) dimension from indicator-valued to real-valued functions.

**Definition 2.1.** *Let $\mathcal{H}$ be a class of real-valued functions with domain $\mathcal{X}$. Let $X_n = \{x_1, ..., x_n\} \subset \mathcal{X}$, and consider a collection of real numbers $s_1, ..., s_n \subset \mathbb{R}^n$. When evaluated at each $x_i$, a function $h \in \mathcal{H}$ will lie on one side[1] of the corresponding $x_i$, i.e. $\mathrm{sign}(h(x_i) - s_i) = \pm 1$. The vector of such sides $(\mathrm{sign}(h(x_i) - s_i))_{i=1}^n$ is thus an element of $\{\pm 1\}^n$.*

*We say that $\mathcal{H}$ P-shatters $X_n$ if there exist real numbers $s_1, ..., s_n$ such that all possible sign combinations are obtained, i.e.,*

$$\{(\mathrm{sign}(h(x_i) - s_i))_i \mid h \in \mathcal{H}\} = \{\pm 1\}^n.$$

*The pseudo-dimension $\dim_p(\mathcal{H})$ is the cardinality of the largest set that is P-shattered:*

$$\dim_p(\mathcal{H}) = \sup \{n \in \mathbb{N} \mid \exists \{x_1, ..., x_n\} \subset \mathcal{X} \text{ that is P-shattered by } \mathcal{H}\}. \tag{1}$$

---

[1] We adopt the notation of $\mathrm{sign}(0) = +1$ for well-definedness, but the other option is equally valid.

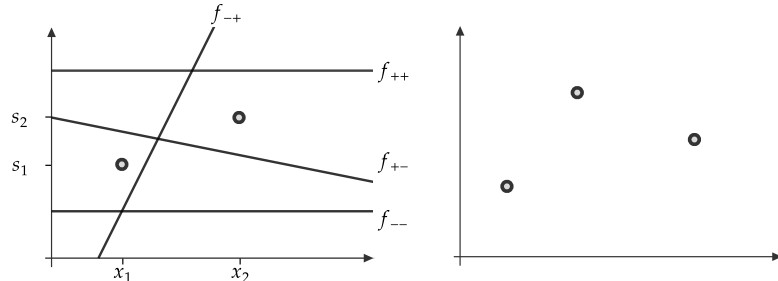

Figure 1: For the class of affine 1D functions $\{x \mapsto ax + b \mid a, b \in \mathbb{R}\}$, this choice of $\{x_1, x_2\} \subset \mathbb{R}$ and $s_1, s_2 \in \mathbb{R}$ on the left is P-shattered. We exhibit affine functions $f_{\pm\pm}$ that take all possible combinations of being above/below $s_i$ at the $x_i$. However, there is no possible arrangement of three points that is P-shattered by affine functions. For example, the arrangement on the right would not have a function that goes below the left and right points but above the middle point. Therefore, the pseudo-dimension of the class of affine 1D functions is 2.

We note that the classical definition of the VC dimension takes a similar form, but without the biases $s_i$ and with $\mathcal{H}$ being a class of binary functions taking values in $\{\pm 1\}$. The pseudo-dimension satisfies similar properties as the VC dimension, such as coinciding with the standard notion of dimension for vector spaces of functions.

**Proposition 2.2** (Anthony & Bartlett 1999, Thm. 11.4). *If $\mathcal{H}'$ is an $\mathbb{R}$-vector space of real-valued functions, then $\dim_P(\mathcal{H}') = \dim(\mathcal{H}')$ as a vector space. In particular, if $\mathcal{H}$ is a subset of a vector space $\mathcal{H}'$ of real-valued functions, then $\dim_P(\mathcal{H}) \leq \dim(\mathcal{H}')$.*

Much like the VC dimension and other statistical complexity quantities such as Rademacher complexity or Gaussian complexity, low complexity leads to better generalization properties of empirical risk minimizers (Bartlett & Mendelson, 2002). One example is as follows, where a precise definition of sample complexity can be found in Appendix A.

**Proposition 2.3** (Anthony & Bartlett 1999, Thm. 19.2). *Let $\mathcal{H}$ be a class of functions mapping from a domain $X$ into $[0, 1] \subset \mathbb{R}$, and that $\mathcal{H}$ has finite pseudo-dimension. Then the $(\epsilon, \delta)$-sample complexity (Definition A.2) is bounded by*

$$m_L(\epsilon, \delta) \leq \frac{128}{\epsilon^2} \left( 2 \dim_P(\mathcal{H}) \log\left(\frac{34}{\epsilon}\right) + \log\left(\frac{16}{\delta}\right) \right). \tag{2}$$

To compare the approximation of one function class by another, we consider a nonlinear width induced by a normed space.

**Definition 2.4** (Nonlinear $n$-width). *Let $\mathcal{F}$ be a normed space of functions. Given two subsets $F_1, F_2 \subset \mathcal{F}$, the (asymmetric) Hausdorff distance between the two subsets is the largest distance between elements of $F_1$ and their closest element in $F_2$:*

$$\text{dist}(F_1, F_2; \mathcal{F}) = \sup_{f_1 \in F_1} \inf_{f_2 \in F_2} \|f_1 - f_2\|_{\mathcal{F}}. \tag{3}$$

*For a subset $F \subset \mathcal{F}$, the nonlinear $n$-width is given by the optimal (asymmetric) Hausdorff distance between $F$ and $\mathcal{H}^n$, infimized over classes $\mathcal{H}^n$ in $\mathcal{F}$ with $\dim_p(\mathcal{H}^n) \leq n$:*

$$\rho_n(F, \mathcal{F}) := \inf_{\mathcal{H}^n} \text{dist}(F, \mathcal{H}^n; \mathcal{F}) = \inf_{\mathcal{H}^n} \sup_{f \in F} \inf_{h \in \mathcal{H}^n} \|f - h\|_{\mathcal{F}}. \tag{4}$$

This width measures the complexity in terms of how closely the entire function class can be approximated with another class of finite pseudo-dimension. This is useful in cases where $F$ has infinite pseudo-dimension, and the nonlinear $n$-width acts as a surrogate measure of complexity, given by how well $F$ can be approximated by classes of finite pseudo-dimension. In Section 3, we provide a lower bound on the nonlinear $n$-width of a bounded Sobolev class of functions. In terms of neural network approximation, these lower bounds complement existing approximation results of ReLU networks, which effectively provide an upper bound on the width by using the class of (bounded width, layers and parameters) ReLU networks as the finite pseudo-dimension approximating class.

## 2.2 RIEMANNIAN GEOMETRY

The manifold hypothesis can be readily expressed in terms of Riemannian geometry. A quick review with definitions is given in Appendix B, and we refer to Bishop & Crittenden (2011); Gallot et al. (2004) for a more detailed exposition.

Throughout, we will assume that our Riemannian manifold is complete, compact, without boundary, and connected. We note that the connectedness assumption can be dropped by working instead on each connected component, since the arguments will be intrinsic and do not depend on any embeddings.

We state the celebrated Bishop–Gromov theorem (Petersen, 2006; Bishop, 1964). This is an essential volume-comparison theorem, used to tractably bound the volume of balls as they grow.

**Theorem 2.5** (Bishop–Gromov). *Let $(M, g)$ be a complete $d$-dimensional Riemannian manifold whose Ricci curvature is bounded below by $\mathrm{Ric} \geq (d-1)K$, for some $K \in \mathbb{R}$. Let $M_K^d$ be the complete $d$-dimensional simply connected space of constant sectional curvature $K$, i.e. a $d$-sphere of radius $1/\sqrt{K}$, $d$-dimensional Euclidean space, or scaled hyperbolic space if $K > 0$, $K = 0$, $K < 0$ respectively. Then for any $p \in M$ and $p_K \in M_K^d$, we have that*

$$\phi(r) = \mathrm{vol}_M(B_r(p))/\mathrm{vol}_{M_K^d}(B_r(p_K)) \tag{5}$$

*is non-increasing on $(0, \infty)$. In particular, $\mathrm{vol}_M(B_r(p)) \leq \mathrm{vol}_{M_K^d}(B_r(p_K))$.*

We note that in a space of constant sectional curvature $M_K^d$, the volume of a ball of radius $r$ does not depend on the center. We thus write $\mathrm{vol}_{M_K^d}(B_r)$ to mean $\mathrm{vol}_{M_K^d}(B_r(p_K))$ for any point $p_K \in M_K^d$ without loss of generality. Bishop–Gromov can be specialized in terms of integrals in hyperbolic space.

**Corollary 2.6** (Block et al. 2020; Ohta 2014). *Let $(M, g)$ be a complete $d$-dimensional Riemannian manifold such that $\mathrm{Ric} \geq (d-1)K$ for some $K < 0$. For any $0 < r < R$, we have*

$$\frac{\mathrm{vol}_M(B_R(x))}{\mathrm{vol}_M(B_r(x))} \leq \frac{\int_0^R s^{d-1}}{\int_0^r s^{d-1}}, \quad s(u) = \sinh(u\sqrt{|K|}). \tag{6}$$

We additionally need the following definitions, bounded in Proposition C.1 and Lemma C.2.

**Definition 2.7** (Packing number). *For a metric space $(M, d)$ and radius $\varepsilon > 0$, the packing number $N_\varepsilon(M)$ is the maximum number of points $x_1, ..., x_n \in M$ such that the open balls $B_\varepsilon(x_i)$ are disjoint. The $\varepsilon$-metric entropy $\mathcal{M}_\varepsilon$ is the maximum number of points $x_1, ..., x_m \in M$ such that $d(x_i, x_j) \geq \varepsilon$ for $i \neq j$.*

**Remark 2.8.** *The following inequality holds:*

$$\mathcal{M}_{2\varepsilon} \leq N_\varepsilon \leq \mathcal{M}_\varepsilon. \tag{7}$$

*The first inequality holds since any $2\varepsilon$-separated subset induces disjoint open $\varepsilon$-balls. The second inequality holds since any set of disjoint $\varepsilon$-balls has necessarily $\varepsilon$-separated centers.*

## 2.3 SOBOLEV FUNCTIONS ON MANIFOLDS

We now define the bounded Sobolev ball on manifolds, which will be the subject of approximation in the next section. There are different ways to define the Sobolev spaces on manifolds due to the curvature differing only by constants, and we consider the variant presented in Hebey (2000).

**Definition 2.9** (Hebey 2000, Sec. 2.2). *Let $(M, g)$ be a smooth Riemannian manifold. For integer $k$, $p \geq 1$, and smooth $u : M \to \mathbb{R}$, define by $\nabla^k u$ the $k$'th covariant derivative of $u$, and $|\nabla^k u|$ its norm, defined in a local chart as*

$$|\nabla^k u|^2 = g^{i_1 j_1}...g^{i_k j_k}(\nabla^k u)_{i_1...i_k}(\nabla^k u)_{j_1...j_k}, \tag{8}$$

*using Einstein's summation convention where repeated indices are summed. Define the set of admissible test functions (with respect to the volume measure) as*

$$\mathcal{C}^{k,p}(M) := \left\{ u \in \mathcal{C}^\infty(M) \mid \forall j = 0, ..., k, \int_M |\nabla^j u|^p \, \mathrm{dvol}_M < +\infty \right\}, \tag{9}$$

*and for $u \in \mathcal{C}^{k,p}(M)$, the Sobolev $W^{k,p}$ norm as*

$$\|u\|_{W^{k,p}} := \sum_{j=0}^{k} \left( \int_M |\nabla^j u|^p \, \mathrm{dvol}_M \right)^{1/p} = \sum_{j=0}^{k} \|\nabla^j u\|_p \tag{10}$$

*The Sobolev space $W^{k,p}(M)$ is defined as the completion of $\mathcal{C}^{k,p}$ under $\|\cdot\|_{W^{k,p}}$.*

It can be shown that Sobolev functions on a compact Riemannian manifold share similar embedding properties as in Euclidean space. We briefly mention the manifold versions of the embedding theorems and Morrey's inequality, which embed into $L^p$ spaces and Hölder spaces respectively. Other corresponding inclusions such as Rellich–Kondrachov and Sobolev–Poincaré also continue to hold, and we refer to Hebey (2000); Aubin (2012) for a more detailed treatment of such results.

We adopt the following definition of a bounded Sobolev ball. This is a natural extension of an $L^p$-ball to Sobolev spaces and provides a compact space of functions to approximate.

**Definition 2.10.** *For constant $C \geq 0$, the bounded Sobolev ball $W^{k,p}(C; M)$ is given by the set of all functions with covariant derivatives bounded in $L^p = L^p(M, \mathrm{vol}_M)$ by $C$:*

$$W^{k,p}(C; M) = \left\{ u \in W^{k,p}(M) \mid \forall l \leq k, \, \|\nabla^l u\|_p \leq C \right\} \tag{11}$$

*We write $W^{k,p}(C)$ to mean $W^{k,p}(C; M)$ for ease of notation.*

## 3 MAIN RESULT

This section begins with a statement of the main approximation result, a lower bound on the nonlinear $n$-width (4) of bounded Sobolev balls $W^{1,p}(1)$ in $L^q$. This is followed by a high-level intuition behind the proof, then the proof in detail. The supporting lemmas are deferred to Appendix C.

**Theorem 3.1.** *Let $(M, g)$ be a $d$-dimensional compact (separable) Riemannian manifold without boundary. From compactness, there exist real constants $K$, $\mathrm{inj}(M)$ such that:*

   1. *The Ricci curvature satisfies $\mathrm{Ric} \geq (d-1)K$, where $K < 0$;*

   2. *The injectivity radius is positive, $\mathrm{inj}(M) > 0$.*

*For any $1 \leq p, q \leq +\infty$, the nonlinear width of $W^{1,p}(1)$ satisfies the lower bound for sufficiently large $n$:*

$$\rho_n(W^{1,p}(1), L^q(M)) \geq C(d, K, \mathrm{vol}(M), p, q)(n + \log n)^{-1/d}. \tag{12}$$

*The constant is independent of any ambient dimension that $(M, g)$ may be embedded in.*

Note that this statement does not refer to any ambient dimension or embeddings, and can be defined on abstract manifolds. This theorem should be contrasted with Maiorov & Ratsaby (1999, Thm. 1), which exhibits a similar bound for the bounded Sobolev space on the unit hypercube $[0, 1]^d$, but with lower bound $n^{-1/d}$. The additional $\log n$ term is necessary due to the curvature of the space. We also note that it is possible to perform this analysis in the case of positive curvature and derive better bounds.

There are two major difficulties in converting the proof of Maiorov & Ratsaby (1999) to the manifold setting, both arising from curvature. Firstly, Maiorov & Ratsaby (1999) uses a partition into hypercubes to construct the desired counterexample. As such hypercube partitions generally do not pose nice properties on manifolds, this must be loosened to a packing of geodesic balls, which does not fully cover the manifold and loosens the bound. The second major difference is the lack of global information, particularly for geodesic balls of the same radius which can have drastically different volumes at different points, even if small-ball volumes behave asymptotically Euclidean pointwise. This will introduce additional constants into the final bound. We break the proof down into the following steps.

### 3.1 PROOF SKETCH

**Step 1.** We consider a class of simple functions, defined as sums of cutoff functions with disjoint supports. The class of simple functions is such that the $L^1$-norm of each component is large within the class of bounded $W^{1,p}(1)$ functions.

**Step 2.** An appropriate subset of the simple functions is then taken, that is isometric to the hyper-cube graph $\{\pm 1\}^m$. Since we can find an $\ell_1$-well separated subset of the hypercube graph, there exists an $L^1$-well separated subset of our simple functions.

**Step 3.** We then show that the $L^1$ separation of the constructed set prevents approximation with any class of insufficiently large pseudo-dimension. This step uses an exponential lower bound on the metric entropy and a polynomial upper bound from Bishop–Gromov to derive a contradiction.

**Step 4.** We conclude that the optimal approximation with function classes with bounded pseudo-dimension must incur an error, bounded from below as in the theorem statement. We conclude the proof by combining all the inequalities from the lemmas and Step 3.

### 3.2 Proof of Theorem 3.1

In the following, $L^p$ spaces will be on $(M, g)$ with respect to the underlying volume measure. Recall the definition of the bounded class of $W^{1,p}$ functions:

$$W^{1,p}(C) = \left\{ u \in W^{1,p}(M) \mid \|u\|_{L^p}, \|\nabla u\|_{L^p} \leq C \right\}. \tag{13}$$

**Step 1. Defining the base function class**. Fix a radius $0 < r < \mathrm{inj}(M)$, which will be chosen appropriately later. Fix a maximal packing of geodesic $r$-balls, say with centers $p_1, ...., p_{N_r}$, where $N_r = N_r^{\mathrm{pack}}(M)$ is the packing number. By definition, $B_r(p_i)$ are disjoint for $i = 1, ..., N_r$. From Proposition C.1, the packing number satisfies the following where $D = \mathrm{diam}(M)$:

$$\frac{\mathrm{vol}(M)}{\mathrm{vol}_{M_K^d}(B_{2r})} \leq N_r^{\mathrm{pack}} \leq \frac{\mathrm{vol}_{M_K^d}(B_D)}{\mathrm{vol}_{M_K^d}(B_r)}. \tag{14}$$

For each ball $B_r(p_i)$, we can construct a $\mathcal{C}^\infty$ function $\phi_i' : M \to [0, r/4]$ with support $\mathrm{supp}(\phi_i') \subset B_r(p_i)$ such that

$$\phi_i'(p) = \begin{cases} r/4, & d(p, p_i) \leq r/2; \\ 0, & d(p, p_i) \geq r. \end{cases} \tag{15}$$

This can be done by constructing a cutoff function and finding an appropriate smooth approximation for separable Riemannian manifolds, using infimal convolutions and $\mathcal{C}^\infty$ partitions of unity (Azagra et al., 2007, Cor. 3). In particular, we can choose $\phi_i'$ to have $|\nabla \phi_i'| \leq 1$ pointwise. From (15), we have the $L^1$ lower bound

$$\|\phi_i'\|_1 \geq (r/4)\mathrm{vol}_M(B_{r/2}(p_i)), \tag{16}$$

Moreover, we have the $L^p$ bounds on $\phi_i'$ and $\nabla \phi_i'$

$$\|\phi_i'\|_p \leq (r/4)\mathrm{vol}_M(B_r(p_i))^{1/p}, \tag{17}$$

$$\|\nabla \phi_i'\|_p \leq \mathrm{vol}_M(B_r(p_i))^{1/p}. \tag{18}$$

Therefore, for $r < 4$, we have that $\phi_i' \in W^{1,p}(\mathrm{vol}_M(B_r(p_i))^{1/p})$. Defining

$$\phi_i := \frac{\phi_i'}{\mathrm{vol}_M(B_r(p_i))^{1/p}}, \tag{19}$$

we get a non-negative function $\phi_i$ with support in $B_r(p_i)$ satisfying:

$$\|\phi_i\|_1 \geq (r/4)\frac{\mathrm{vol}_M(B_{r/2}(p_i))}{\mathrm{vol}_M(B_r(p_i))^{1/p}}, \quad \phi_i \in W^{1,p}(1). \tag{20}$$

Moreover, $\phi_i = r/(4\mathrm{vol}_M(B_r(p_i))^{1/p})$ on $B_{r/2}(p_i)$. We now consider the function class

$$F_r = \left\{ f_a = \frac{1}{N_r^{1/p}} \sum_{i=1}^{N_r} a_i \phi_i \,\middle|\, a_i \in \{\pm 1\}, i = 1, ..., N_r \right\}. \tag{21}$$

Since the sum is over functions of disjoint support, we have that $\|f_a\|_p, \|\nabla f_a\|_p \leq 1$, and thus each element of $F_r$ also lies in $W^{1,p}(1)$. Moreover, every element $f_a \in F_r$ satisfies the $L^1$ lower bound using (20):

$$\|f_a\|_1 \geq \frac{r}{4N_r^{1/p}} \sum_{i=1}^{N_r} \frac{\text{vol}_M(B_{r/2}(p_i))}{\text{vol}_M(B_r(p_i))^{1/p}}, \quad \forall f_a \in F_r. \tag{22}$$

**Step 2.** $L^1$-**well-separation of** $F_r$**.** Consider the following lemma, which shows the existence of a large well-separated subset of $\ell_1^m$.

**Lemma 3.2** (Lorentz et al. 1996, Lem. 2.2)**.** *There exists a set $G \subset \{\pm 1\}^m$ of cardinality at least $2^{m/16}$ such that for any $v \neq v' \in G$, the distance $\|v - v'\|_{\ell_1^m} \geq m/2$. In particular, any two elements differ in at least $m/4$ entries.*

In particular, let $G \subset \{\pm 1\}^{N_r}$ be well separated by the above lemma. Denote by $F_r(G)$ the subset of $F_r$ corresponding to these indices:

$$F_r(G) = \{f_a \in F_r \mid a \in G\}. \tag{23}$$

For the specific choice of separated $G \subset \{\pm 1\}^{N_r}$ in the above lemma, we claim the following well-separation of $F_r(G)$, proved in Appendix D.

**Claim 1.** *There exists a constant $C_1(r) > 0$ such that for any $f \neq f' \in F_r(G)$, we have*

$$\|f - f'\|_1 \geq C_1(r) > 0. \tag{24}$$

*Moreover, the following constant works:*

$$C_1(r) = \frac{rN_r^{1-1/p} \int_0^{r/2} s^{d-1}}{8 \int_0^r s^{d-1}} \inf_{i \in [N_r]} \left[ \text{vol}_M(B_r(p_i))^{1-1/p} \right]. \tag{25}$$

This shows $L^1$-well separation of the subset $F_r(G) \subset F_r \subset W^{1,p}(1)$, which consist of sums of disjoint cutoff functions. The key will now be to contrast this with the metric entropy bounds in Lemma C.2, by showing that $F_r(G)$ is difficult to approximate with function classes of low pseudo-dimension.

**Step 3a. Construction of well-separated bounded set.** Let $\mathcal{H}^n$ be a given set of $\text{vol}_M$-measurable functions with $\dim_p(\mathcal{H}^n) \leq n$. Let $\varepsilon > 0$. Denote

$$\delta = \sup_{f \in F_r(G)} \inf_{h \in \mathcal{H}^n} \|f - h\|_1 + \varepsilon = \text{dist}(F_r(G), \mathcal{H}^n, L^1(M)) + \varepsilon. \tag{26}$$

Define a projection operator $P : F_r(G) \to \mathcal{H}^n$, mapping any $f \in F_r(G)$ to any element $Pf$ in $\mathcal{H}^n$ such that

$$\|f - Pf\|_1 \leq \delta. \tag{27}$$

We introduce a (measurable) clamping operator $\mathcal{C}$ for a function $f$:

$$\beta_i = r/(4\text{vol}_M(B_r(p_i))^{1/p}N_r^{1/p}), \quad i = 1, ..., N_r, \tag{28}$$

$$(\mathcal{C}f)(x) = \begin{cases} -\beta_i, & x \in B_r(p_i) \text{ and } f(x) < -\beta_i; \\ f(x), & x \in B_r(p_i) \text{ and } -\beta_i \leq f(x) \leq \beta_i; \\ \beta_i, & x \in B_r(p_i) \text{ and } f(x) > \beta_i; \\ 0, & \text{otherwise.} \end{cases} \tag{29}$$

Note that $\beta_i$ are the bounds of $f_a \in F_r$ in the balls $B_r(p_i)$. Now consider the set of functions $\mathcal{S} := \mathcal{C}PF_r(G)$. Suppose $f \neq f' \in F_r(G)$. We show separation in $\mathcal{S}$ using triangle inequality:

$$\|\mathcal{C}Pf - \mathcal{C}Pf'\|_1 \geq \|f - f'\|_1 - \|f - \mathcal{C}Pf\|_1 - \|f' - \mathcal{C}Pf'\|_1. \tag{30}$$

For any $a \in G$, we have that $f_a \leq \beta_i$ in $B_r(p_i)$, and both $f_a$ and $\mathcal{C}Pf_a$ are zero on $M \setminus \bigsqcup_i B_r(p_i)$. We thus have that for any $x \in M$ and any $f_a \in F_r(G)$,

$$|f_a(x) - \mathcal{C}Pf_a(x)| \leq |f_a(x) - Pf_a(x)|. \tag{31}$$

This inequality holds for $x \in B_r(p_i)$ since $\mathcal{C}$ clamps $Pf_a(x)$ towards $[-\beta_i, \beta_i]$, and holds trivially on $M \setminus \bigsqcup_i B_r(p_i)$. Integrating and using (27), we have that for any $f_a \in F_r(G)$,

$$\|f_a - \mathcal{C}Pf_a\|_1 \leq \|f_a - Pf_a\|_1 \leq \delta. \tag{32}$$

Using (24), (30) and (32), we thus have separation

$$\|\mathcal{C}Pf - \mathcal{C}Pf'\|_1 \geq \|f - f'\|_1 - 2\delta \geq C_1(r) - 2\delta. \tag{33}$$

**Step 3b. Minimum distance by contradiction.** Suppose for contradiction that $\delta \leq C_1(r)/4$. Then from (33), we have

$$\|\mathcal{C}Pf - \mathcal{C}Pf'\|_1 \geq C_1(r)/2. \tag{34}$$

In particular, the separation implies that the $\mathcal{C}Pf$ are distinct for distinct $f \in F_r(G)$, thus $|\mathcal{S}| = |G| \geq 2^{N_r/16}$.

Define $\alpha = C_1(r)/2$. Consider the metric entropy in $L^1$, as given in Lemma C.2. By construction (33), $\mathcal{S}$ itself is an $\alpha$-separated subset in $L^1$ as any two elements are $L^1$-separated by $\alpha$, so

$$\mathcal{M}_\alpha(\mathcal{S}, L^1(\text{vol}_M)) \geq 2^{N_r/16}. \tag{35}$$

We now wish to obtain an upper bound on $\mathcal{M}_\alpha(\mathcal{S}, L^1)$ using Lemma C.2. From the definition of pseudo-dimension, we have $\dim_p(\mathcal{C}PF_r(G)) \leq \dim_p(PF_r(G))$, since any P-shattering set for $\mathcal{C}PF_r(G)$ will certainly P-shatter $PF_r(G)$. Since $PF_r(G) \subset \mathcal{H}^n$, we have $\dim_p(PF_r(G)) \leq \dim_p(\mathcal{H}^n) \leq n$. Thus $\dim_p(\mathcal{S}) = \dim_p(\mathcal{C}PF_r(G)) \leq n$. $\mathcal{S}$ is $L^1$-separated with distance at least $\alpha$, and moreover consists of elements that are bounded by $\beta := \sup_i \beta_i$. Lemma C.2 now gives:

$$M_\alpha(\mathcal{S}, L^1(\text{vol}_M)) \leq e(n+1) \left( \frac{4e\beta\text{vol}(M)}{\alpha} \right)^n. \tag{36}$$

Intuitively, $N_r \sim r^{-d}$, so the lower bound (35) is exponential in $r$. Meanwhile, $\alpha$ and $\beta$ are both polynomial in $r$, so the upper bound (36) is polynomial in $r$. So for sufficiently small $r$, we have a contradiction with the supposition that $\delta \leq C_1(r)/4$. We now show this formally. Recall:

$$\beta = \sup_{i \in [N_r]} \frac{r}{4\text{vol}_M(B_r(p_i))^{1/p} N_r^{1/p}}, \quad \alpha = \frac{rN_r^{1-1/p} \int_0^{r/2} s^{d-1}}{16 \int_0^r s^{d-1}} \inf_{i \in [N_r]} \left[ \text{vol}_M(B_r(p_i))^{1-1/p} \right]. \tag{37}$$

Note that the supremum in $\beta$ and the infimum in $\alpha$ is attained by the same[2] $i \in [N_r]$, namely, the $p_i$ that has smallest $\text{vol}_M(B_r(p_i))$. Combining (35) and (36), where $s(u) = \sinh(u\sqrt{|K|})$,

$$2^{N_r/16} \leq e(n+1) \left( \frac{4e\beta\text{vol}(M)}{\alpha} \right)^n$$

$$= e(n+1) \left( \frac{4e\text{vol}(M) \sup_{i \in [N_r]} [r/(4\text{vol}_M(B_r(p_i))^{1/p} N_r^{1/p})]}{\frac{rN_r^{1-1/p} \int_0^{r/2} s^{d-1}}{16 \int_0^r s^{d-1}} \inf_{i \in [N_r]} \left[ \text{vol}_M(B_r(p_i))^{1-1/p} \right]} \right)^n$$

$$= e(n+1) \left( 16e \frac{\text{vol}(M) \int_0^r s^{d-1}}{N_r \int_0^{r/2} s^{d-1}} \sup_{i,j \in [N_r]} \frac{\text{vol}_M(B_r(p_j))^{1/p-1}}{\text{vol}_M(B_r(p_i))^{1/p}} \right)^n$$

$$= e(n+1) \left( 16e \frac{\text{vol}(M) \int_0^r s^{d-1}}{N_r \int_0^{r/2} s^{d-1}} \sup_{i \in [N_r]} \left[ \text{vol}_M(B_r(p_i))^{-1} \right] \right)^n \tag{38}$$

where the equalities come from definition of $\beta$ and $\alpha$ and rearranging, and the last equality from noting the supremum is attained when $i = j \in [N_r]$ minimizes $\text{vol}_M(B_r(p_i))$. The following result lower-bounds the volume of small balls to control the supremum term.

---

[2]Intuitively, tall thin functions have the worst $L^1$ to $L^p$ ratio compared to short fat functions. By construction, small balls have tall thin functions.

**Proposition 3.3** (Croke 1980, Prop. 14). *For $r \leq \mathrm{inj}(M)/2$, the volume of the ball $B_r(p)$ satisfies*

$$\mathrm{vol}_M(B_r(p)) \geq C_2(d)r^d, \quad C_2(d) = \frac{2^{d-1}\mathrm{vol}_{M_1^{d-1}}(B_1)^d}{d^d\mathrm{vol}_{M_1^d}(B_1)^{d-1}}. \tag{39}$$

The volume of the $d$-dimensional hyperbolic sphere with sectional curvature $K$ can be written in terms of the volume of the $d$-dimensional sphere[3]:

$$\mathrm{vol}_{M_K^d}(B_\rho) = \mathrm{vol}_{M_1^d}(B_1) \int_{t=0}^{\rho} \left( \frac{\sinh(\sqrt{|K|}t)}{\sqrt{|K|}} \right)^{d-1} \mathrm{d}t. \tag{40}$$

Note that $x \leq \sinh(x) \leq 2x$ for $x \in [0,2]$. Therefore, for $\rho \leq 2/\sqrt{|K|}$, we have $\sinh(\sqrt{|K|}t) \leq 2\sqrt{|K|}t$. We thus have that

$$\begin{aligned}
\mathrm{vol}_{M_K^d}(B_\rho) &= \mathrm{vol}_{M_1^d}(B_1) \int_{t=0}^{\rho} \left( \frac{\sinh(\sqrt{|K|}t)}{\sqrt{|K|}} \right)^{d-1} \mathrm{d}t \\
&< \mathrm{vol}_{M_1^d}(B_1)2^{d-1}\rho^d/d = C_3(d)\rho^d, \quad C_3(d) := \mathrm{vol}_{M_1^d}(B_1)2^{d-1}/d.
\end{aligned} \tag{41}$$

Moreover,

$$\frac{\int_0^r s^{d-1}}{\int_0^{r/2} s^{d-1}} = \frac{\int_0^r \sinh(\sqrt{|K|}u)^{d-1}\,\mathrm{d}u}{\int_0^{r/2} \sinh(\sqrt{|K|}u)^{d-1}\,\mathrm{d}u} \leq 2^d \quad \text{for } r < 1/\sqrt{K}. \tag{42}$$

We continue the inequality (38) for $r < 1/\sqrt{|K|}$:

$$2^{N_r/16} \leq e(n+1) \left( 16e\frac{\mathrm{vol}(M)\int_0^r s^{d-1}}{N_r \int_0^{r/2} s^{d-1}} \sup_{i \in [N_r]} \left[ \mathrm{vol}_M(B_r(p_i))^{-1} \right] \right)^n \tag{43}$$

$$\leq e(n+1) \left( 16e\,\mathrm{vol}_{M_K^d}(B_{2r})\frac{\int_0^r s^{d-1}}{\int_0^{r/2} s^{d-1}} C_2^{-1}r^{-d} \right)^n \qquad \text{using (14), Prop. 3.3} \tag{44}$$

$$\leq e(n+1) \left( 16eC_3(2r)^d\frac{\int_0^r s^{d-1}}{\int_0^{r/2} s^{d-1}} C_2^{-1}r^{-d} \right)^n \qquad \text{using (41)} \tag{45}$$

$$\leq e(n+1) \left( 2^{d+4}eC_3\frac{\int_0^r s^{d-1}}{\int_0^{r/2} s^{d-1}} C_2^{-1} \right)^n \tag{46}$$

$$\leq e(n+1) \left( 2^{2d+4}eC_3C_2^{-1} \right)^n = e(n+1)C_4(d)^n, \qquad \text{using (42)} \tag{47}$$

where

$$C_4 = C_4(d) := 2^{2d+4}eC_3C_2^{-1} = 2^{2d+4}ed^{d-1} \left[ \frac{\mathrm{vol}_{M_1^d}(B_1)}{\mathrm{vol}_{M_1^{d-1}}(B_1)} \right]^d. \tag{48}$$

We get a contradiction if

$$N_r > 16 \left[ n\log_2 C_4 + \log_2\left(e(n+1)\right) \right]. \tag{49}$$

Recalling the lower bound (14) on $N_r$ and using (42),

$$N_r \geq \frac{\mathrm{vol}(M)}{\mathrm{vol}_{M_K^d}(B_{2r})} > \frac{\mathrm{vol}(M)}{C_3(2r)^d}. \tag{50}$$

Take the following choice of $r$:

$$r = \min\left\{ \frac{1}{2} \left( 16\frac{C_3}{\mathrm{vol}(M)} \left[ n\log_2 C_4 + \log_2(e(n+1)) \right] \right)^{-1/d}, \frac{1}{\sqrt{|K|}}, \frac{\mathrm{inj}(M)}{2}, 4 \right\}. \tag{51}$$

---

[3]The volume of the $d$-dimensional sphere is $2\pi^{d/2}/\Gamma(d/2)$, where $\Gamma$ is Euler's gamma function.

Using (50), this choice of $r$ satisfies the contradiction condition (49). Note $r \sim (n + \log n)^{-1/d}$. The constants $C_3, C_4$ depend only on $d$.

**Step 4. Concluding contradiction.** This choice of $r$ contradicts the assumption that $\delta \leq C_1(r)/4$. Therefore, we must have that $\delta > C_1/4$. Since the choice of $r$ is independent of the choice of $\varepsilon > 0$ taken at the start of Step 3a, we have that

$$\text{dist}(F_r(G), \mathcal{H}^n, L^1(\text{vol}_M)) \geq C_1(r)/4, \tag{52}$$

where $r$ is chosen as in (51). We obtain the chain of inequalities

$$\begin{aligned}
\text{dist}(W^{1,p}(1), \mathcal{H}^n, L^q) &\geq \text{dist}(W^{1,p}(1), \mathcal{H}^n, L^1)\text{vol}(M)^{1/q-1} \\
&\geq \text{dist}(F_r(G), \mathcal{H}^n, L^1)\text{vol}(M)^{1/q-1} \\
&\geq C_1(r)\text{vol}(M)^{1/q-1}/4 \\
&= \frac{rN_r^{1-1/p} \int_0^{r/2} s^{d-1}}{32 \int_0^r s^{d-1}} \inf_{i \in [N_r]} \left[ \text{vol}_M(B_r(p_i))^{1-1/p} \right] \text{vol}(M)^{1/q-1}, \tag{53}
\end{aligned}$$

where the first inequality comes from Hölder's inequality $\|u\|_1 \leq \|u\|_q \text{vol}(M)^{1-1/q}$, the second inequality from $F_r(G) \subset W^{1,p}(1)$, the third from (52) and the equality from definition (25) of $C_1(r)$. We conclude with recalling the bounds (42), (50), and Proposition 3.3. We have

$$\text{dist}(W^{1,p}(1), \mathcal{H}^n, L^q) \geq \frac{r}{32} \underbrace{\left( \frac{\text{vol}(M)}{C_3(2r)^d} \right)^{1-1/p}}_{(50)} \underbrace{2^{-d}}_{(42)} \underbrace{\left[ C_2 r^d \right]^{1-1/p}}_{\text{Prop. 3.3}} \text{vol}(M)^{1/q-1}$$

$$= C_5(d, \text{vol}(M), p, q)r.$$

The constant is

$$C_5 = 2^{-d-5} \frac{\text{vol}(M)^{1/q-1/p}}{2^{d-d/p}} (C_2 C_3^{-1})^{1-1/p}. \tag{54}$$

Moreover, the constant $C_5$ and choice of $r$ are independent of $\mathcal{H}^n$. Taking infimum over all choice of $\mathcal{H}^n$ with $\dim_P(\mathcal{H}^n) \leq n$ and using (51), we have

$$\rho_n(W^{1,p}(1), L^q) \geq C_5(d, \text{vol}(M), p, q)r \gtrsim (n + \log n)^{-1/d}. \tag{55}$$

$\square$

## 4 CONCLUSION

This work provides a theoretical motivation to further explore the manifold hypothesis. We show that the problem of approximating a bounded class of Sobolev functions depends only on the intrinsic properties of the space it is supported in. More precisely, the approximation error of the bounded $W^{1,p}$ space with respect to bounded pseudo-dimension classes is shown to be at least $(n + \log n)^{-1/d}$, where $d$ is the intrinsic dimension of the underlying manifold. Since generalization error is linear in pseudo-dimension, this provides an ambient-dimension-free lower-bound on generalization error. This is in contrast to many works in the literature that provide constructive upper bounds on generalization error based on ReLU approximation properties that still depend on the embedding of the manifold in ambient Euclidean space. Followup work could consider alternative statistical complexities, such as Rademacher or Gaussian complexity.

The proposed bound can be improved in multiple ways. Firstly, the analysis is restricted to one weak derivative. The analogous result of approximating $W^{k,p}$ in the cube $[0, 1]^D$ has lower bounds $\sim n^{-k/D}$ (Maiorov & Ratsaby, 1999). Extending our analysis to more weak derivatives would require a careful construction in Step 1 of test functions with the appropriate regularity conditions. In particular, we would require cutoff functions $\text{supp}\,\phi_i \subset B_r(p_i)$ with explicitly bounded higher weak derivatives $\|\nabla^\alpha \phi_i\| \lesssim r^{|\alpha|}$, which do not seem to appear in the literature. The explicit construction of Moulis (1971) of a $\mathcal{C}^\infty$ function that approximates a $\mathcal{C}^{2k-1}$ function in the $\mathcal{C}^k$-topology could be useful in this. Moreover, the current bound requires knowledge of the injectivity radius to uniformly lower-bound the volume of small balls. Other ways of constructing volume lower-bounds would help in improving the constants in the bound.

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

# A  SAMPLE COMPLEXITY

For completeness, we briefly formalize the sample complexity bound Proposition 2.3, based on (Anthony & Bartlett, 1999).

**Definition A.1** (Anthony & Bartlett 1999, Def. 16.4). *For a set of functions $F$, an* approximate sample error minimizing *(approximate-SEM) algorithm $\mathcal{A}(z, \epsilon)$ takes any finite number of samples $z = (x_i, y_i)_{i=1}^m$ in $\cup_{m=1}^\infty (X \times \mathbb{R})^m$ and an error bound $\epsilon > 0$, and outputs an element $f \in F$ satisfying*

$$\mathcal{R}(\mathcal{A}(F, z)) < \inf_{f \in F} \mathcal{R}(f) + \epsilon, \quad \mathcal{R}_m(f) = \frac{1}{m} \sum_{i=1}^m (f(x_i) - y_i)^2. \tag{56}$$

**Definition A.2** (Anthony & Bartlett 1999, Def. 16.1). *For a set of functions $F$ mapping from domain $X$ to $[0, 1]$, a* learning algorithm $L$ for $F$ *is a function taking any finite number of samples,*

$$L : \bigcup_{m=1}^\infty (X \times \mathbb{R})^m \to F \tag{57}$$

*with the following property. For any $\epsilon, \delta \in (0, 1)$, there is an integer (*sample complexity*) $m_0(\epsilon, \delta)$ such that if $m \geq m_0(\epsilon, \delta)$, the following holds for any probability distribution $P$ on $X \times [0, 1]$.*

*If $z$ is a training sample of length $m$ according to the product distribution $P^m$ (i.i.d. samples), then with probability at least $1 - \delta$, the function $L(z)$ output by $L$ is such that*

$$\mathbb{E}_{(x,y) \sim P}[L(z)(x) - y]^2 < \inf_{f \in F} \mathbb{E}_{(x,y) \sim P}[f(x) - y]^2 + \epsilon. \tag{58}$$

*In other words, given $m \geq m_0$ training samples, the squared-risk of the learning algorithm's output is $\epsilon$-optimal with probability at least $1 - \delta$.*

Observe that the approximate-SEM algorithm works on the empirical risk, while the learning algorithm works on the risk. Relating the two thus gives generalization bounds. The formal version of Proposition 2.3, based on (Anthony & Bartlett, 1999) is now given as follows.

**Proposition A.3** (Anthony & Bartlett 1999, Thm. 19.2). *Let $\mathcal{H}$ be a class of functions mapping from a domain $X$ into $[0, 1] \subset \mathbb{R}$, and that $\mathcal{H}$ has finite pseudo-dimension. Let $\mathcal{A}$ be any approximate-SEM algorithm, and define for samples $z$, $L(z) = \mathcal{A}(z, 16/\sqrt{length(z)})$. Then $L$ is a learning algorithm for $\mathcal{H}$, and its sample complexity is bounded as follows:*

$$m_L(\epsilon, \delta) \leq \frac{128}{\epsilon^2} \left( 2 \dim_P(\mathcal{H}) \log\left(\frac{34}{\epsilon}\right) + \log\left(\frac{16}{\delta}\right) \right). \tag{59}$$

## A.1  RELATION TO OUR BOUNDS

In computing a risk minimizer over the Sobolev ball, we need to make two practical simplifications: namely parameterizing the Sobolev ball (into a function class of finite pseudo-dimension), and in simplifying the risk from (typically) an expectation into an empirical version. Our bound targets the former approximation, while the aforementioned sample complexity bounds targets the latter generalization problem. To formalize this, we have the relationship between the three quantities:

$$\underset{f \in W^{1,p}(1)}{\arg\min} \mathcal{R}(f) \longleftrightarrow \underset{f \in \mathcal{H}_n}{\arg\min} \mathcal{R}(f) \longleftrightarrow \underset{f \in \mathcal{H}_n}{\arg\min} \mathcal{R}_m(f),$$

where $\mathcal{H}_n$ is some function class with pseudo-dimension at most $n$. For the sake of exposition, we make some additional assumptions and work in the worst-case.

Suppose that the (expected) risk $\mathcal{R} : W^{1,p}(1) \to \mathbb{R}$ is Lipschitz continuous in the space of functions, such as $\mathcal{R} = \mathbb{E}_\mu[\|f(x)-y\|^2]$ for some sufficiently regular probability measure $\mu \in \mathcal{P}(\mathcal{M} \times \mathbb{R})$ and measurement space $\mathcal{Y}$. Consider the set of minimizers $\mathcal{G} := \arg\min_{f \in W^{1,p}(1)} \mathcal{R}(f)$, and assume that $\arg\min_{f \in W^{1,p}(1)} \mathcal{R}(f) = \arg\min_{f \in L^q} \mathcal{R}(f)$, i.e., risk minimizers in $L^q$ are also in $W^{1,p}$.

Take $\mathcal{H}_n \subset L^q$ to be an optimal approximating class of pseudo-dimension at most $n$. In the worst case, we have that the $L^q$ distance between $\mathcal{H}_n$ and minimizers $g \in \mathcal{G}$ is bounded from below by some $\epsilon = \epsilon(n) > 0$ as given by Theorem 3.1. Assuming further some strong convexity conditions, this gives that the minimizer $f^* \in \arg\min_{f \in \mathcal{H}_n} \mathcal{R}(f)$ has risk at least $c\epsilon$ for some strong-convexity constant $c$. Adding this worst-case risk with the worst-case risk of the $(\epsilon, \delta)$-sample complexity bounds, we have that an empirical risk minimizer may have even greater risk.

We make these assumptions for the sake of exposition in the worst-case; note however that Theorem 3.1 considers the furthest element in $W^{1,p}(1)$ from $\mathcal{H}_n$, and that minimizers in $\mathcal{H}_n$ and $W^{1,p}(1)$ may be closer together.

## B RIEMANNIAN GEOMETRY

**Definition B.1.** *A $d$-dimensional* Riemannian manifold *is a real smooth manifold $M$ equipped with a Riemannian metric $g$, which defines an inner product on the tangent plane $T_pM$ at each point $p \in M$. We assume $g$ is smooth, i.e. for any smooth chart $(U, x)$ on $M$, the components $g^{ij} = g(\frac{\partial}{\partial x_i}, \frac{\partial}{\partial x_j}) : U \to \mathbb{R}$ are $\mathcal{C}^\infty$.*

*A manifold is without boundary if every point has a neighborhood homeomorphic to an open subset of $\mathbb{R}^d$. For a point $p \in M$, let $B_r(p)$ be the metric ball around $p$ in $M$ with radius $r > 0$.*

*The* sectional (or Riemannian) curvature *takes at each point $p \in M$, a tangent plane $P \subset T_pM$ and outputs a scalar value. The* Ricci curvature *(function) $\mathrm{Ric}(v) \equiv \mathrm{Ric}(v, v)$ of a unit vector $v \in T_pM$ is the mean sectional curvature over planes containing $v$ in $T_pM$. In particular, for a manifold with constant sectional curvature $K$, we have $\mathrm{Ric} \equiv (d-1)K$. We write $\mathrm{Ric} \geq K$ for $K \in \mathbb{R}$ to mean that $\mathrm{Ric}(v) \geq K$ holds for all unit vectors in the tangent bundle $v \in TM$.*

*The* injectivity radius $\mathrm{inj}(p)$ *at a point $p \in M$ is the supremum over radii $r > 0$ such that the exponential map defines a global diffeomorphism (nonsingular derivative) from $B_r(0; T_pM)$ onto its image in $M$. The injectivity radius $\mathrm{inj}(M)$ of a manifold is the infimum of such injectivity radii over all points in $M$.*

*A Riemannian manifold has a (unique) natural volume form, denoted $\mathrm{vol}_M$. In local coordinates, the volume form is*

$$\mathrm{vol}_M = \sqrt{|g|}\, \mathrm{d}x_1 \wedge ... \wedge \mathrm{d}x_d, \tag{60}$$

*where $g$ is the Riemannian metric, and $\mathrm{d}x_1, ..., \mathrm{d}x_d$ is a (positively-oriented) cotangent basis. We drop the subscripts when taking the volume of the whole manifold $\mathrm{vol}(M) = \mathrm{vol}_M(M)$.*

Intuitively, the sectional curvature controls the behavior of geodesics that are close, and the Ricci curvature controls volumes of small balls. For manifolds of positive sectional curvature such as on a sphere, geodesics tend to converge, and small balls have less volume than Euclidean balls. In manifolds with negative sectional curvature such as hyperbolic space, geodesics tend to diverge, and small balls have more volume than Euclidean balls.

Within the ball of injectivity, geodesics are length-minimizing curves. The injectivity radius defines the largest ball on which the geodesic normal coordinates may be used, where it locally behaves as $\mathbb{R}^d$. This is an intrinsic quantity of the manifold, which does not depend on the embedding.

The volume form can be thought of as a higher-dimensional surface area, where the scaling term $\sqrt{|g|}$ arises from curvature and choice of coordinates. For example, for the 2-sphere $\mathbb{S}^2$ embedded in $\mathbb{R}^3$, the volume form is simply the surface measure, which can be expressed in terms of polar coordinates.

From the compactness assumption, we have that the sectional (and hence Ricci) curvature is uniformly bounded from above and below (Bishop & Crittenden, 2011, Sec. 9.3), and the injectivity radius $\mathrm{inj}(M)$ is positive (Cheeger et al., 1982; Grant, 2012).

**Proposition B.2.** *Let $(M, g)$ be a compact Riemannian manifold without boundary. The following statements hold.*

1. *(Bounded curvature) The sectional curvature (and hence the Ricci curvature) is uniformly bounded from above and below (Bishop & Crittenden, 2011, Sec. 9.3).*

2. *(Positive injectivity radius) Let the sectional curvature $K_m$ be bounded by some $|K_M| \leq K$. Suppose there exists a point $p \in M$ and constant $v_0 > 0$ such that $\mathrm{vol}_M(B_1(p)) \geq v_0$. Then there exists a positive constant $i_1 = i_1(K, v_0, d)$ such that (Cheeger et al., 1982; Grant, 2012):*
$$\mathrm{inj}(p) \geq i_1 > 0$$
*In particular, since $M$ is compact, using a finite covering argument, $\mathrm{inj}(M)$ is bounded below by some positive constant.*

## C    PACKING LEMMAS

**Proposition C.1** (Packing number estimates)**.** *Suppose $(M, g)$ has curvature lower-bounded by $K \in \mathbb{R}$, diameter $D$ and dimension $d$. Let $M_K^d$ be the $d$-dimensional model space of constant sectional curvature $K$ (i.e. sphere, Euclidean space, or hyperbolic space). The packing number $N_\varepsilon(M)$ satisfies, where $p_K$ is any point in $M_K^d$:*

$$\frac{\mathrm{vol}(M)}{\mathrm{vol}_{M_K^d}(B_{2\varepsilon})} \leq N_\varepsilon \leq \frac{\mathrm{vol}_{M_K^d}(B_D)}{\mathrm{vol}_{M_K^d}(B_\varepsilon)} \tag{61}$$

*Proof.* Let $\{p_1, ..., p_{N_\varepsilon}\}$ be an $\varepsilon$-packing of $M$.

**Lower bound.** By maximality, balls of radius $2\varepsilon$ at the $p_i$ cover $M$, so we have by summing over volumes and using Bishop–Gromov:

$$\mathrm{vol}(M) \leq \sum_{i=1}^{N_\varepsilon} \mathrm{vol}_M(B_{2\varepsilon}(p_i)) \leq N_\varepsilon \mathrm{vol}_{M_K^d}(B_{2\varepsilon}). \tag{62}$$

**Upper bound.** Apply Theorem 2.5 with $\varepsilon \leq D$. We have $\mathrm{vol}_M(B_\varepsilon(p)) \leq \mathrm{vol}_{M_K^d}(B_\varepsilon(p_K))$ and $\mathrm{vol}_M(B_D(p_i)) = \mathrm{vol}(M)$ for all $i$. Since the $\varepsilon$-balls are disjoint, we have by finite additivity and Bishop–Gromov:

$$\mathrm{vol}_M(M) \geq \sum_{i=1}^{N_\varepsilon} \mathrm{vol}_M(B_\varepsilon(p_i)) \geq N_\varepsilon \mathrm{vol}(M) \frac{\mathrm{vol}_{M_K^d}(B_\varepsilon)}{\mathrm{vol}_{M_K^d}(B_D)}. \tag{63}$$

$\square$

We additionally consider a bound on the metric entropy for bounded functions.

**Lemma C.2** (Haussler 1995, Cor. 2 and 3)**.** *For any set $X$, any probability distribution $P$ on $X$, any distribution $Q$ on $\mathbb{R}$, any set $\mathcal{F}$ of $P$-measurable real-valued functions on $X$ with $\dim_P(\mathcal{F}) = n < \infty$ and any $\varepsilon > 0$, the $\varepsilon$-metric entropy $\mathcal{M}_\varepsilon$ (largest cardinality of a $\varepsilon$-separated subset, where distance between any two elements is $\geq \varepsilon$) satisfies:*

$$\mathcal{M}_\varepsilon(\mathcal{F}, \sigma_{P,Q}) \leq e(n+1) \left( \frac{2e}{\varepsilon} \right)^n. \tag{64}$$

*Specifically, taking $L^1$ distance, if $\mathcal{F}$ is $P$-measurable taking values in the interval $[0, 1]$, we have*

$$\mathcal{M}_\varepsilon(\mathcal{F}, L^1(P)) \leq e(n+1) \left( \frac{2e}{\varepsilon} \right)^n. \tag{65}$$

*If $\sigma$ is instead a finite measure, and $\mathcal{F}$ is $\sigma$-measurable taking values in the interval $[-\beta, \beta]$, then*

$$\mathcal{M}_\varepsilon(\mathcal{F}, L^1(\sigma)) \leq e(n+1) \left( \frac{4e\beta\sigma(X)}{\varepsilon} \right)^n. \tag{66}$$

**Remark C.3.** *The final inequality comes from the second-to-last inequality, by noting that a $\varepsilon$-separated set in $\sigma$ corresponds to a $\varepsilon/\sigma(X)$-separated set in the normalized measure $\sigma/\sigma(X)$, as well as scaling everything by $2\beta$.*

# D    SEPARATION CLAIM

Here we show Claim 1 . Recall that $G \subset \{\pm 1\}^{N_r}$ is defined to be well-separated by Lemma 3.2.

**Claim.** *For any $f \neq f' \in F_r(G)$, we have*

$$\|f - f'\|_1 \geq \frac{r N_r^{1-1/p} \int_0^{r/2} s^{d-1}}{8 \int_0^r s^{d-1}} \inf_{i \in [N_r]} \left[ \mathrm{vol}_M(B_r(p_i))^{1-1/p} \right]. \tag{67}$$

*Proof.* Suppose $f \neq f' \in F_r(G)$. In particular, they are generated by multi-indices $a \neq a' \in G$. Consider the set of indices $\mathcal{I} \subset [N_r]$ such that $a_i \neq a'_i$. By construction in Lemma 3.2, $|\mathcal{I}| \geq N_r/4$. Then the difference between $f$ and $f'$ on $B_r(p_i)$ is $2\phi_i/N_r^{1/p}$ if $i \in \mathcal{I}$, and 0 otherwise. By disjointness of the $B_r(p_i)$, we have

$$\|f - f'\|_1 = \sum_{i \in \mathcal{I}} \frac{2}{N_r^{1/p}} \|\phi_i\|_1 \geq \frac{r}{2 N_r^{1/p}} \sum_{i \in \mathcal{I}} \frac{\mathrm{vol}_M(B_{r/2}(p_i))}{\mathrm{vol}_M(B_r(p_i))^{1/p}} \tag{68}$$

$$\geq \sum_{i \in \mathcal{I}} \frac{r \int_0^{r/2} s^{d-1} \mathrm{vol}_M(B_r(p_i))}{2 N_r^{1/p} \int_0^r s^{d-1} \mathrm{vol}_M(B_r(p_i))^{1/p}} \qquad \text{where } s(u) = \sinh(u\sqrt{|K|}) \tag{69}$$

$$\geq \frac{r N_r^{1-1/p} \int_0^{r/2} s^{d-1}}{8 \int_0^r s^{d-1}} \inf_{i \in \mathcal{I}} \left[ \mathrm{vol}_M(B_r(p_i))^{1-1/p} \right] \tag{70}$$

$$\geq \frac{r N_r^{1-1/p} \int_0^{r/2} s^{d-1}}{8 \int_0^r s^{d-1}} \inf_{i \in [N_r]} \left[ \mathrm{vol}_M(B_r(p_i))^{1-1/p} \right] \tag{71}$$

by the $L^1$-bound on $\phi_i$, Bishop–Gromov (Corollary 2.6), and using $|I| \geq N_r/4$ for the inequalities respectively. $\qquad\square$

