# OpenReview forum: "Blessing of Dimensionality for Approximating Sobolev Classes on Manifolds"
_ICLR.cc/2025/Conference — Submitted to ICLR 2025_

### Official Review · Reviewer_StSY · 2024-11-01

**Soundness:** 4
**Presentation:** 3
**Contribution:** 2
**Rating:** 5
**Confidence:** 3

**Summary:**

This paper is focused on the manifold assumption in machine learning. The goal is to further shed light on how the intrinsic dimension of the data manifold enters into notions of complexity in function approximation. In particular, the authors prove lower bounds on the complexity of approximating Sobolev space functions on manifolds, and show that the lower bounds, which are essentially 1/n^(1/d), depend only on the intrinsic dimension d of the manifold, and not the ambient dimension of the space the manifold lies in. The authors use a notion of pseudodimension that is an extension of VC-dimension and measure complexity by the nonlinear n-width.

**Strengths:**

The question of why deep neural networks work well on extremely high dimensional data is an important problem and the manifold hypothesis may be a good way to explain this. Work on this problem is important and valuable in machine learning. The problem of lower bounds on complexity is not studied as often as upper bounds. The results appear to be new and non-trivial.

**Weaknesses:**

It's not clear to me how applicable these results are in practice. Even when data satisfies the manifold assumption, the intrinsic dimension d may be quite large. It is not clear how large the authors think d is in practice, and how large a d would make these results applicable vs vacuous. For MNIST, for example, it's often given that d is between 7 and 14, depending on the digit. One can assume d is much larger for more challenging problems, maybe 20-40? In this case, the error bound 1/n^(1/d) is vacuous, unless the number of data points n is astronomically large (e.g., if d=20 we need 10^(20) data points!).

**Questions:**

I don't understand why the main result, Theorem 3.1, is measuring nonlinear n-width between the Sobolev space W^{1,p} and the Lebesgue space L^q. It's really unclear to me why this implies anything about the difficulty in approximating Sobolev space functions. I'd like to see this clarified.

---

> ### Author Response · Authors · 2024-11-21
> **Response to Reviewer StSY**
>
> We thank the reviewer for the kind review. Please find point-by-point responses to the concerns and comments below.
>
> > It's not clear to me how applicable these results are in practice. Even when data satisfies the manifold assumption, the intrinsic dimension d may be quite large. It is not clear how large the authors think d is in practice, and how large a d would make these results applicable vs vacuous. For MNIST, for example, it's often given that d is between 7 and 14, depending on the digit. One can assume d is much larger for more challenging problems, maybe 20-40? In this case, the error bound $1/n^{1/d}$ is vacuous, unless the number of data points n is astronomically large (e.g., if d=20 we need $10^{20}$ data points!).
>
> This work is intended as an extension of classical theory into the manifold hypothesis setting. Existing approximation results are equally vacuous for standard imaging examples, and moreover, estimates of intrinsic dimension vary wildly between methods. However, from the independence of ambient dimension, practical guidance could include minimizing the intrinsic dimensionality by reducing the number of uninformative variates. Our results quantitatively show that such variates increase the pseudodimension required to uniformly approximate the Sobolev ball with an exponential factor.
>
> We would like to clarify that in Theorem 3.1, $n$ refers to the pseudo-dimension with which we are allowed to approximate the Sobolev class. As a concrete example, pseudodimension of ReLU neural networks scales as $\Omega(W \log W)$, where $W$ is the number of parameters, see for example Theorem 3 in [Bartlett2019]. The dependence on data points comes afterwards, using e.g. Proposition 2.3, albeit giving a sample complexity that is linear in the allowed pseudodimension. Towards the gap between the theoretical dataset requirements and practical machine learning, this could be because the Sobolev class is still a very descriptive class of functions. Rates for more structured classes of functions could be cause for future work.
>
> > I don't understand why the main result, Theorem 3.1, is measuring nonlinear n-width between the Sobolev space $W^{1,p}$ and the Lebesgue space $L^q$. It's really unclear to me why this implies anything about the difficulty in approximating Sobolev space functions. I'd like to see this clarified.
>
> The main result concerns the approximation of the unit Sobolev ball using function classes of finite pseudo-dimension, where the approximation distance is measured in terms of maximum $L^q$ distance. Since the unit Sobolev ball has infinite pseudo-dimension, approximation with finite complexity classes is necessary for computation. We provide a lower bound for how much complexity is required to ($L^q$-uniformly) approximate functions in the unit Sobolev ball, which can be informally translated as how complicated the Sobolev ball is.
>
> ### References
> [Bartlett2019] Bartlett, P. L., Harvey, N., Liaw, C., \& Mehrabian, A. (2019). Nearly-tight VC-dimension and pseudodimension bounds for piecewise linear neural networks. Journal of Machine Learning Research, 20(63), 1-17.

---

> > ### Comment · Reviewer_StSY · 2024-11-27
> >
> > Thanks for the clarifications. I have decided to leave my score as is, since I feel the paper is somewhat limited in applicability.

---

### Official Review · Reviewer_qm9Z · 2024-11-03

**Soundness:** 2
**Presentation:** 2
**Contribution:** 2
**Rating:** 5
**Confidence:** 3

**Summary:**

The paper concerns a lower bound for a nonlinear width (involving the pseudo-dimension, a generalized version of the VC dimension) of Sobolev classes over smooth manifolds of dimension $d$. The authors claim that while the manifold can be embedded in a higher dimensional space with dimesion $D \gg d$ the width of the Sobolev class has a lower bound that depends only on the dimension $d$ of the manifold.

**Strengths:**

For a technical paper, the presentation is approachable and is self-contained (modulo some typos and missing definitions). The paper extends the lower bound proved in (Maiorov and Ratsaby, 1999) to manifolds.

**Weaknesses:**

- Not only is the result of Theorem 1 independent of ambient dimension $D$, the ambient dimension does not appear _anywhere_ in the estimates. This is somewhat odd because the abstract mentions the manifold hypothesis which concerns both $D$ and $d$. In some similar approximation bounds, there is typically some dependence on $D$. The authors should address this.

- In a similar vein, the authors do not present a connection between the sample complexity mentioned in Proposition 2.3 and the main Theorem 1, as far as I can see. The assumed connection is that, due to this property of classes with finite pseudo-dimension $\mathcal{H}_n$, the Sobolev class can also be estimated with the sample complexity given in (2), once the approximating class $\mathcal{H}_n$ is determined, it can be estimated with this sample complexity. This connection should be made somewhere.

- The main structure of the proof is almost identical to (Maiorov and Ratsaby, 1999), except the construction of the $L^1$-separated set of functions, due to the domain being a manifold. There are a few questions about the extended lower bound, which I ask below in the "questions" section.

**Questions:**

- The Theorem 1 lower bound (12) has a dependence on $p$, unlike in the Euclidean case (Maiorov and Ratsaby, 1999). Is there a plausible reason for this, i.e. is $p$ there for an inherent reason?

- Why is Theorem 1 restricted to the case $K < 0$? What is different about the positive curvature case?

- line 535: it is mentioned that the cutoff functions with bounds on higher derivatives is difficult to construct, but I am having trouble seeing why this should be so. Can the authors explain further?

- line 758: The authors say "By maximality, balls of radius $2\epsilon$ at the $p_i$ cover $M$" but why is this true? The manifold can potentially be very narrow.

- line 691: What is the notation $\mathcal{A}(z, 16 / \sqrt{\text{length}(z)})$?

- Should the empirical risk in (56) be scaled by the sample size?

---

> ### Author Response · Authors · 2024-11-21
> **Response to Reviewer qm9Z**
>
> We thank the reviewer for the detailed review. The authors would be happy to add additional background into the appendix if the reviewer believes that it would aid the readability of the paper. The typos have been fixed in the revised paper. Please additionally find below point-by-point responses to the reviewer's concerns.
>
> > Not only is the result of Theorem 1 independent of ambient dimension $D$, the ambient dimension does not appear anywhere in the estimates. This is somewhat odd because the abstract mentions the manifold hypothesis which concerns both $D$ and $d$. In some similar approximation bounds, there is typically some dependence on $D$. The authors should address this.
>
> Existing approximation results typically consider explicit constructions to construct upper bounds on the statistical complexity. Such constructions generally come in terms of neural networks, in which the dependence on the ambient dimension comes naturally as a function of the output dimension. In this work, we consider instead an abstract manifold without any extrinsic structure as given by (isometric) embeddings into $\mathbb{R}^d$. As such, there is no dependence on the ambient dimension $D$ in our results.
>
> > In a similar vein, the authors do not present a connection between the sample complexity mentioned in Proposition 2.3 and the main Theorem 1, as far as I can see. The assumed connection is that, due to this property of classes with finite pseudo-dimension $\mathcal{H}_n$, the Sobolev class can also be estimated with the sample complexity given in (2), once the approximating class $\mathcal{H}_n$ is determined, it can be estimated with this sample complexity. This connection should be made somewhere.
>
> Indeed, our result provides a worst-case lower bound on the approximation error, assuming that the true solution lies in an area that is not well approximated by $\mathcal{H}_n$. There is a tradeoff between the approximation between the risk minimizers in the Sobolev ball and in $\mathcal{H}_n$, as well as the generalization error incurred within $\mathcal{H}_n$, with the former decreasing and latter increasing as the allowed pseudo-dimension $n$ increases. Our result then characterizes the first tradeoff, while the second is given by classical generalization bounds/statistical complexity results. We add a short section to discuss this, left to the appendix due to space limitations in the main text.
>
> > The main structure of the proof is almost identical to (Maiorov and Ratsaby, 1999), except the construction of the $L^1$-separated set of functions, due to the domain being a manifold. There are a few questions about the extended lower bound, which I ask below in the "questions" section.
>
> We address the reviewer's questions below. We note that in addition to the generalization from the unit hypercube $[0,1]^D$ to a general (compact separable Riemannian) manifold without boundary, our result also gives the exact dimension dependence by giving explicit constants, further extending (Maiorov and Ratsaby, 1999).
>
> > The Theorem 1 lower bound (12) has a dependence on $p$, unlike in the Euclidean case (Maiorov and Ratsaby, 1999). Is there a plausible reason for this, i.e. is $p$ there for an inherent reason?
>
> The dependence arises from the fact that the volume of a manifold is not identically 1, as the Euclidean case considers only the unit hypercube $[0,1]^d$. The dependence on $p$ comes from an application of H\"older's inequality to bound the constructions in their respective spaces, and are always attached to a $\mathrm{vol}(M)$.
>
> > Why is Theorem 1 restricted to the case $K<0$? What is different about the positive curvature case?
>
> The estimate will be tighter in the positive curvature case, as volume estimates will be tighter. We present the result for the case where $K<0$ since this is more general. The proof can be easily modified for $K>0$ by changing Equation (40) to the corresponding form for elliptic space, namely replacing $\sinh(\sqrt{|K|t})$ with $\sin(\sqrt{Kt})$ (for $\rho < \pi \sqrt{|K|}$) to derive the same result, but with a different requirement for $r$ in Equation (51).
>
> > line 535: it is mentioned that the cutoff functions with bounds on higher derivatives is difficult to construct, but I am having trouble seeing why this should be so. Can the authors explain further?
>
> While intuitively simple in the Euclidean case, going further than one derivative introduces Riemann curvature terms that must be individually controlled (corresponding to curvature when computing cross derivatives), such as in Equation (8). The authors are unaware of similar constructions in the literature that explicitly uniformly control the higher-order covariant derivatives. While a uniform bound on the Riemann curvature tensor $g^{ij}$ would allow for this, we consider only bounded Ricci curvature as it is more standard in the literature.

---

> > ### Author Response · Authors · 2024-11-21
> > **Response to Reviewer qm9Z continued**
> >
> > > line 758: The authors say "By maximality, balls of radius $2\epsilon$ at the $p_i$ cover $M$" but why is this true? The manifold can potentially be very narrow.
> >
> > Suppose balls of radius $2\epsilon$ did not cover $M$. Then there exists a point $p$ of $M$ which is of distance at least $2\epsilon$ away from all the $p_i$. Then, open balls of radius $\epsilon$ centered at the points $\{p_1,...,p_{N_\epsilon},p\}$ are all disjoint, contradicting maximality of the packing by definition.
> >
> > > line 691: What is the notation $\mathcal{A}(z, 16/\sqrt{\textrm{length}(z)}$?
> >
> > There are some typos in Definition A.1 which have now been fixed. $\mathcal{A}(z, \epsilon)$ takes a set of samples $z$ and an error bound $\epsilon$, and produces an approximate solution to the empirical risk/sample error minimization problem. Proposition A.3 can be interpreted as the learning algorithm also requiring the approximate empirical risk/sample error minimizing oracle $\mathcal{A}$ to be increasingly precise as the sample size increases. These have now been fixed in the revised version.
> >
> > > Should the empirical risk in (56) be scaled by the sample size?
> >
> > Yes, fixed.

---

> > > ### Comment · Reviewer_qm9Z · 2024-12-02
> > >
> > > Thank you for the answers. I will keep the current score.

---

### Official Review · Reviewer_vHxw · 2024-11-04

**Soundness:** 3
**Presentation:** 3
**Contribution:** 2
**Rating:** 6
**Confidence:** 4

**Summary:**

This paper studies the complexity of Sobolev function class on Riemannian manifolds. Specifically, the paper derives lower bound of the approximation error of a Sobolev ball by a smaller class with complexity bounded as pseudo-dimension. By constructing explicitly functions of bounded Sobolev norm that are separated in $L^1$, the paper connects the packing number of the manifold with a hard-to-learn subclass in the Sobolev ball, thus forcing a larger error/width. The main theorem claims a lower bound that only depends on intrinsic quantities of the manifold.

**Strengths:**

The paper has a natural motivation, and the concluded rate seems matching that of classical Euclidean case. The presentation is lucid, and the proof sketch and extended discussion is well written. Overall this paper is a solid contribution on the topic of manifold learning.

**Weaknesses:**

1. The result is not too surprising on the 1,p-Sobolev class, and considering that higher Sobolev space can even be an RKHS [1], one would expect major improvement on the rate. Also using volume comparison to control the packing number is rather standard, and one might further ask if the same technique is applicable to metric measure spaces or RCD spaces, though I understand this technicality may not be particularly befitting of this venue.


2. Typos:
Definition 2.7 is defining packing number not covering number, and also metric entropy is not commonly defined this way. This version of metric entropy is exactly the same as packing number, hence (7) is not needed. (The key proposition C.1 is correct.) $CPf_a$ out side the balls on line 375 is not necessarily 0.

[1] De Vito, Ernesto, Nicole Mücke, and Lorenzo Rosasco. "Reproducing kernel Hilbert spaces on manifolds: Sobolev and diffusion spaces." Analysis and Applications 19.03 (2021): 363-396.

**Questions:**

See above.

---

> ### Author Response · Authors · 2024-11-21
> **Response to Reviewer vHxw**
>
> We thank the reviewer for the kind review. Please find point-by-point responses to the reviewer's concerns.
>
> > The result is not too surprising on the 1,p-Sobolev class, and considering that higher Sobolev space can even be an RKHS [1], one would expect major improvement on the rate. Also using volume comparison to control the packing number is rather standard, and one might further ask if the same technique is applicable to metric measure spaces or RCD spaces, though I understand this technicality may not be particularly befitting of this venue.
>
> Indeed, we recover rates as given by Euclidean intuition. We note that since we derive rates in negatively curved space, the rates are slightly worse than Euclidean. Conversely, in positively curved space, one may be able to derive better constants using more precise bounds when using Bishop--Gromov. Asymptotically however, one should expect the same rates as the Euclidean case, as the neighborhoods given by packings informally become flat.
>
> Faster rates require higher regularity of $W^{k,p}$, which change the lower bound from $n^{-1/D}$ to $n^{-k/D}$ in the Euclidean case. We note that construction of functions with appropriately bounded higher derivatives is more difficult than the Euclidean case due to the presence of curvature, and the authors are not aware of a result similar to [Azagra2007]. While the authors believe this is possible, this could introduce additional superexponential dependence on $d$, and require a uniform bound on the Riemann curvature tensor instead of only the Ricci curvature.
>
> We speculate from a brief look at [Sturm2006] that the result may still hold in RCD spaces given appropriate analogs of Bishop--Gromov, as well as a uniform control on the measure of small balls (Prop. 3.3). However, the authors are not familiar with this line of work.
>
> > Typos: Definition 2.7 is defining packing number not covering number, and also metric entropy is not commonly defined this way. This version of metric entropy is exactly the same as packing number, hence (7) is not needed. (The key proposition C.1 is correct.) $\mathcal{C}Pf_a$ out side the balls on line 375 is not necessarily 0.
>
> Changed the incorrect naming from covering number to packing number (Def. 2.7). We believe there are separate communities that define metric entropy differently (namely log of the covering number). We have replaced "metric entropy" with "$\epsilon$-metric entropy" in Definition 2.7 and Lemma C.2, as there does not seem to be an alternative name for the cardinality of the largest $\epsilon$-separated subset.
>
> By definition in Equation (29), $\mathcal{C}$ clamps any function to 0 outside the balls $B_r(p_i)$.
>
> ### References
> [Sturm2006] Sturm, Karl-Theodor. ``A curvature-dimension condition for metric measure spaces." Comptes Rendus Mathematique 342.3 (2006): 197-200.
>
> [Azagra2007] Daniel Azagra, Juan Ferrera, Fernando Lopez-Mesas, and Yenny Rangel. Smooth approximation of Lipschitz functions on Riemannian manifolds. Journal of Mathematical Analysis and Applications, 326(2):1370–1378, 2007.

---

> > ### Comment · Reviewer_vHxw · 2024-11-21
> >
> > I thank the authors for the response, which addressed my questions on generalizations to several directions. I thus decide to leave the score unchanged, and raise my confidence to 4.

---

### Author Response · Authors · 2024-11-26

We thank the reviewers for their kind and helpful comments on our theoretical work. We have revised the paper, fixing minor typos and adding a short discussion into the appendix as to how our bounds fit into error decomposition, marked in blue. As the pdf revision deadline is approaching, we would be happy to incorporate any additional small changes that the reviewers may find helpful for our work, as well as answer any outstanding theoretical or conceptual queries. Thank you.

---

### Meta-Review · Area_Chair_s8Xk · 2024-12-25

**Metareview:**

The paper studies the complexity of Sobolev spaces on manifolds. Its main result is a lower bound on the error incurred by approximating Sobolev functions in Lq with subspaces of functions learnable from n samples; the bound scales as roughly n^{-1/d} where d is the dimension of the manifold. Here, the constant depends on d, K (a lower bound on curvature), vol(M), p and q. The proof extends an argument of Maiorov and Ratsaby on functions on [0,1]^d by constructing well separated collections of functions on M — roughly, these are W^{1,p} functions localized to a packing set of geodesic balls. The main modification of the argument is to control the effect of curvature (the main difference vis-a-vis the hypercube). The implication of this result for learning is to provide a lower bound for learning (intrinsic) Sobolev functions which depends only on intrinsic quantities.

This is a technically solid paper, which provides lower bounds on the sample complexity of learning on Riemannian manifolds. The results here are complementary to existing upper bounds for extrinsic learners (such as ReLU networks) on submanifolds. The paper provides lower bounds which are independent of embedding — depending only on the intrinsic properties of M [of course, the sample complexity of extrinsic learners may depend on embedding]. The main concerns pertained to (A) the significance of the technical innovations in the paper, and (B) the practical implications of this setting and results for learning in moderate d.

**Additional Comments On Reviewer Discussion:**

The main points of discussion included
(A) the possibility of extending the results to higher order Sobolev spaces, improving the rate from n^{-1/d} to n^{-k/d}.
(B) the implication of the type of approximation developed here for learning — in particular,
 - the implication on extrinsic learners such as neural networks
 - the relationship between nonlinear n-widths in Lq and learnability
The paper provides lower bounds which are independent of the embedding. As the authors and reviewers both note results on higher order Sobolev spaces may have stronger implications for learnability (since the nonparametric rate of n^{-1/d} is quite slow for d moderate).

---

### Decision · Program_Chairs · 2025-01-22

Reject